# Changes in Medial Elbow Joint Parameters Due to Selective Contraction of the Forearm Flexor–Pronator Muscles

**DOI:** 10.3390/healthcare11040586

**Published:** 2023-02-15

**Authors:** Hiroshi Hattori, Kiyokazu Akasaka, Takahiro Otsudo, Yutaka Sawada, Toby Hall

**Affiliations:** 1School of Physical Therapy, Saitama Medical University, Moroyama 350-0496, Japan; 2Department of Physical Therapy, Saitama Medical University Graduate School of Medicine, Moroyama 350-0495, Japan; 3School of Health Sciences, Tokyo University of Technology, Ota 114-8535, Japan; 4Curtin School of Allied Health, Curtin University, WA 6102, Australia

**Keywords:** medial elbow joint, forearm flexor–pronator muscles, dynamic stabilizers, selective contraction, ultrasound, injury prevention, youth sports, baseball

## Abstract

The harder the forearm flexor–pronator muscles (FPMs) relative to the ulnar collateral ligament (UCL), the less likely it is for UCL laxity to occur with repeated pitching. This study aimed to clarify what selective contraction of the forearm muscles makes FPMs harder relative to UCL. The study evaluated 20 elbows of male college students. Participants selectively contracted the forearm muscles in eight conditions under gravity stress. The medial elbow joint width and the strain ratio indicating tissue hardness of the UCL and FPMs during contraction were evaluated using an ultrasound system. Contraction of all FPMs (in particular flexor digitorum superficialis [FDS] and pronator teres [PT]) decreased the medial elbow joint width compared to rest (*p* < 0.05). UCL hardens due to the contraction involving the FDS muscle (*p* < 0.05). FPMs harden due to the contraction of flexor carpi ulnaris (FCU) and FDS (*p* < 0.05). In the ratio of strain ratio UCL/FPMs, there was no significant difference between rest and each contraction task (*p* > 0.05). However, contractions composed of FCU and PT tended to harden FPMs relative to the UCL. FCU and PT activation may be effective in UCL injury prevention.

## 1. Introduction

Pitching in baseball players is classified into six phases: wind-up phase, stride phase, arm cocking phase, arm acceleration phase, arm deceleration phase, and follow-through phase [1]. In addition, this movement generates a great valgus load in the elbow joint shortly before the arm reaches maximum external rotation [2,3]. The ulnar collateral ligament (UCL) is a primary static stabilizer of the medial elbow joint against approximately 54% of the valgus load [4]. The UCL is burdened when elbow valgus load occurs during each pitching motion; hence, UCL injuries among baseball players are generally considered to be caused by overuse.

In previous studies, Mahure et al. [5] reported that the number of UCL reconstruction surgeries performed between 2003 and 2014 increased by 343%, with an average annual incidence per 100,000 people for ages 15 to 19 years significantly greater than for all other age groups. Furthermore, subsequent projections suggest that incidence in the age groups 15 to 19 years and 20 to 24 years will continue to rapidly increase. Additionally, Conte et al. [6] reported that pitchers have a high prevalence of UCL reconstruction surgery in professional baseball, with 25% of major league pitchers and 15% of minor league pitchers having a history of the surgery. Furthermore, Cain et al. [7] reported that the ratio of players returning to the previous level of competition or higher after UCL reconstruction surgery was only 83%, and the average time to full competition was 11.6 months after UCL reconstruction surgery. These reports indicate that the incidence of elbow joint injuries in baseball players is very high and the impact on players is great. Therefore, the prevention of medial elbow injuries, including UCL injuries, is a continuing issue among baseball players, especially adolescents.

Various studies have reported on pitching biomechanics related to elbow valgus load [8,9,10], the identification of risk factors related to elbow injury [11,12,13,14], pitch count limits [15,16], and injury prevention programs [17,18] for the prevention of elbow injury in baseball players. In recent years, the forearm flexor–pronator muscles (FPMs), which are dynamic stabilizers of the medial elbow joint [19,20], have been considered important for preventing UCL injuries. It has been reported that FPMs are highly active from the arm cocking phase to the arm acceleration phase in the pitching motion [21]. In addition, basic research has shown that contraction of the FPMs decreases the medial elbow joint width (i.e., stabilizes) and hardens and strengthens the medial elbow joint structures [22]. Therefore, it would be necessary for UCL injury prevention to further reduce the contribution rate of the UCL, which is against approximately 54% valgus load [4] during pitching, by activating the FPMs, which are dynamic medial elbow stabilizers.

Several previous studies have evaluated the effect of repeated load on the medial elbow joint due to repeated pitching [23,24,25]. Hattori et al. [23] reported that repeated pitching of 60 pitches in high school baseball players caused an immediate increase in elbow valgus laxity. In addition, Matsuo et al. [24] reported that even in elementary school baseball players and college baseball players, elbow valgus laxity immediately increased after 60 pitches. Furthermore, focusing on the medial elbow stabilizers, which is a soft tissue, Hattori et al. [25] reported that repeated pitching 100 times in high school baseball players immediately caused UCL laxity through repeated load on the UCL. Furthermore, they reported that the degree of UCL laxity during repeated pitching was related to the ratio of the hardness of the UCL and FPMs, and that the harder the FPMs relative to the UCL, the less likely it is for UCL laxity to develop [25], thus highlighting the protective nature of the FPMs in preventing UCL laxity. However, although the factors related to UCL laxity during repeated pitching have been presented, the changes in the medial elbow stabilizers due to the selective contraction of the forearm muscles need further clarification. Furthermore, the effective contraction of the forearm muscles that makes the FPMs harder relative to the UCL has yet to be defined.

Therefore, the purpose of this study targeting general healthy adolescents as a basic study is (1) to evaluate changes in medial elbow joint parameters due to selective contraction of the forearm muscles and (2) to identify what selective contraction of the forearm muscles makes the FPMs harder relative to the UCL. We hypothesized that selective contraction of the FCU running directly above the UCL [26] would make the FPMs harder relative to the UCL. The clarification of these findings will help identify future prevention programs for UCL injuries among baseball players.

## 2. Materials and Methods

### 2.1. Participants

The study evaluated 20 elbows of the dominant arm in 20 healthy male college students (mean [range] ± standard deviation: age, 21.4 [20,21,22] ± 0.8 years; height, 171.1 [163.0–179.0] ± 4.3 cm; weight, 66.3 [57.1–81.9] ± 8.0 kg) who applied to participate. All participants were right-hand dominant. The dominant arm was defined as the arm that players use to throw a ball and was determined by interview. Participants were excluded from the study if they had elbow pain or a history of elbow surgery. None of the applicants for participation were excluded by the exclusion criteria. All participants signed a research consent form prior to data collection. This study was approved by the Ethics Committee of Saitama Medical University, Saitama, Japan (2021-018).

### 2.2. Procedures

Participants were supine on the bed, with the shoulder joint in 90° abduction, the elbow joint in 90° flexion, and the forearm in a neutral position. Participants’ arm positions were verified using a goniometer. Furthermore, the arm to be measured was moved away from the edge of the bed and gravity stress due to the weight of the forearm was applied to the elbow joint to evaluate the parameters of the medial elbow joint. Gravity stress used in this study has been used in many previous studies [23,24,25,27,28], and has been reported to be as useful as the Telos device in evaluating the medial elbow joint width [28]. From its designated position, participants performed resting and seven selective contractions of forearm muscles (Figure 1): (1) resting, (2) wrist ulnar flexion (contraction of flexor carpi ulnaris [FCU]), (3) finger flexion (contraction of flexor digitorum superficialis [FDS]), (4) forearm pronation (contraction of pronator teres [PT]), (5) wrist ulnar flexion and finger flexion (contraction of FCU and FDS), (6) wrist ulnar flexion and forearm pronation (contraction of FCU and PT), (7) finger flexion and forearm pronation (contraction of FDS and PT), (8) wrist ulnar flexion, finger flexion, and forearm pronation (contraction of FCU, FDS, and PT). Participants were asked to perform each muscle contraction task with the maximum voluntary contraction. Each of the seven muscle contraction tasks (other than resting) was performed 3 times at maximum effort for 5 s, and ultrasound measurements were taken during contraction each time. The order of these tasks was randomized. None of the participants had difficulty performing these contraction tasks due to pain or fatigue.

### 2.3. Measurement

An ultrasound imaging system (Noblus; Hitachi Aloka Medical, Tokyo, Japan) and an ultrasound 18–5 MHz linear array transducer (L64; Hitachi Aloka Medical, Tokyo, Japan) were used to measure the medial elbow joint width in B-mode and the strain ratio indicative of tissue hardness in UCL and FPM in real-time ultrasound sonoelastography (Figure 2). In addition, grip strength was measured for subject characteristics. One experienced physical therapist (H.H.) with over 11 years of experience using ultrasound imaging systems performed all measurements.

Ultrasound image extraction was performed by placing the ultrasound transducer over the medial elbow joint to ensure that the medial epicondyle of the humerus, trochlea of the humerus, sublime tubercle of the ulna, UCL, and FPM were covered so that a consistent image was extracted [29]. Ultrasound evaluations were performed based on previous studies by Sasaki et al. [27] and Hattori et al. [25]. Medial elbow joint width was assessed by describing the distal–medial corner of the trochlea of the humerus and the proximal edge of the sublime tubercle of the ulna in B mode and measuring the distance between the two points (ulnohumeral joint width). A greater value in medial elbow joint width indicates greater elbow valgus laxity, while a smaller value indicates less elbow valgus laxity. The examiner performed the process from image extraction to measurement three times. The average of the three trials was then used for data analysis.

Real-time ultrasound sonoelastography colors ultrasound images according to tissue hardness (red, soft; yellow to green, medium; blue, hard) and can measure strain ratio indicative of tissue hardness. Strain ratios indicative of the hardness in the UCL and FPMs were quantified by attaching an acoustic coupler (EZU-TECPL1, Hitachi Aloka Medical, Tokyo, Japan) to the transducer and using the coupler region (just above the target region) as the reference region. Greater strain ratio values indicate harder tissues (stiffer tissues), while less values indicate softer tissues. The UCL measurement region was defined as a circle with the maximum ligament width at the midpoint. The FPMs measurement region was defined as a circle 0.5 cm in diameter at the superficial layer of the FPMs above the ulnohumeral joint. Selection of the measurement region was performed on the B-mode image so that the examiner would not be influenced by the coloring of hardness on the image. The examiner performed the process from image extraction to measurement three times. The average of the three trials was then used for data analysis. The reliability of these assessment methods has been reported as good to excellent (ICC1,3: UCL, 0.91 and 0.83; FPMs, 0.85 and 0.80) [22]. In addition, the ratio of strain ratio UCL/FPMs was calculated from the measured UCL and FPMs data to evaluate the hardness of FPM relative to UCL. A smaller value in the ratio of strain ratio UCL/FPMs indicates the harder tissue of FPMs relative to that of UCL.

Grip strength of the dominant arm was measured using a grip strength meter (GRIP-D T.K.K.5401; Takei Scientific Instruments Co, Ltd., Niigata, Japan) before performing muscle contraction tasks as a physical characteristic of the participants. Measurements of the grip strength were performed by maintaining a gripping motion at maximal effort for 5 s in a standing, drooping upper extremity position with elbows extended [25]. The average of three trials was calculated.

### 2.4. Sample Size

A power analysis was performed a priori to determine the sample size to obtain statistical significance at an appropriate power (1–β) of 80% (G*Power 3.1.9.4, http://www.gpower.hhu.de/, accessed on 1 April 2019). In order to compare each ultrasound parameter across the eight tasks, the effect size in the F-test was set at 0.25 (α = 0.05, 1–β = 0.8). A total of 16 participants were needed for the power analysis. Therefore, 20 participants were recruited.

### 2.5. Statistical Analysis

Measured data were analyzed using SPSS Statistics for Windows (Version 26.0; IBM Corp., Armonk, NY, USA). One-way repeated-measures ANOVA or Friedman’s test were performed to examine the main effect of the muscle contraction tasks on each medial elbow joint parameter. After that, as post hoc tests, Dunnett tests were used to compare rest and muscle contraction tasks. Significant differences were set at a level of 0.05.

## 3. Results

As the physical characteristic of the participants, the grip strength of the participants was 41.0 ± 4.7 kg. The results of each medial elbow joint parameter in eight conditions are shown in Table 1 and Figure 3, Figure 4, Figure 5 and Figure 6.

For the medial elbow joint width, a main effect of the tasks was observed (*p* < 0.001). Medial elbow joint width during tasks other than “FCU alone contractions” were significantly reduced compared to the resting condition (Rest, 4.07 ± 0.65; FDS, 3.42 ± 0.58, *p* = 0.008; PT, 3.40 ± 0.55, *p* = 0.006; FCU and FDS, 3.46 ± 0.63, *p* = 0.017; FCU and PT, 3.48 ± 0.63, *p* = 0.022; FDS and PT, 3.08 ± 0.63, *p* < 0.001; FCU, FDS, and PT, 3.00 ± 0.68, *p* < 0.001).

For the strain ratio of the UCL, a main effect of the tasks was observed (*p* < 0.001). The strain ratio of the UCL in “FDS”, “FCU and FDS,” and “FCU, FDS, and PT” increased significantly compared to the resting condition (Rest, 5.84 ± 5.12; FDS, 23.58 ± 18.68, *p* = 0.017; FCU and FDS, 36.66 ± 30.04, *p* < 0.001; FCU, FDS, and PT, 26.36 ± 31.02, *p* = 0.004).

For the strain ratio of the FPMs, a main effect of the tasks was observed (*p* < 0.001). The strain ratio of the FPMs in “FCU”, “FDS”, “FCU and FDS” and “FCU, FDS, and PT” increased significantly compared to the resting condition (Rest, 0.34 ± 0.18; FCU, 1.20 ± 1.38, *p* = 0.006; FDS, 1.15 ± 0.91, *p* = 0.012; FCU and FDS, 1.60 ± 0.88, *p* < 0.001; FCU, FDS, and PT, 1.13 ± 0.53, *p* = 0.014).

For the ratio of the strain ratio UCL/FPMs, a main effect of the tasks was observed (*p* = 0.013). There was no significant difference between the resting condition and each contraction task (All *p* > 0.05). However, contraction tasks including FDS tended to increase compared to the resting condition, but “FCU”, “PT”, and “FCU and PT” tended to decrease compared to the resting condition.

## 4. Discussion

This study investigated (1) changes in the medial elbow stabilizers due to selective contraction of the forearm muscles, and (2) what selective contraction of the forearm muscles makes the FPMs harder relative to the UCL. Although medial elbow joint width was little changed by FCU contraction alone, it was significantly decreased (i.e., increase stabilization of the medial elbow joint) during activation of FDS and PT alone and when combined. Therefore, it was suggested that FCU contributes little to the stabilization of the medial elbow joint, while FDS and PT greatly contribute to that. From the above, it was suggested that muscle strengthening of the entire FPMs (especially FDS and PT) may be effective for stabilizing the medial elbow joint because the more muscles recruited, the more the medial elbow joint width decreased. The present study’s results that contraction of the FPMs decreases the medial elbow joint width are similar to previous studies [20,22,30]. New findings from this study are the different stabilizing effects of various selective contractions of the forearm muscles against elbow valgus load.

With regard to the effect of contraction of various forearm muscles against elbow valgus load, the hardness of the UCL was significantly greater during tasks involving FDS contraction. On the other hand, the hardness of the FPMs showed little change during PT alone contraction but significantly increased by contraction of FCU and FDS alone or combined. Focusing on the ratio of hardness in the UCL and FPMs, although there was no significant difference in the ratio of the strain ratio UCL/FPMs between the eight conditions, the UCL/FPMs tended to be greater in the tasks involving FDS contraction tasks. On the other hand, the ratio of strain ratio UCL/FPMs tended to be less in contraction tasks composed of FCU and PT, suggesting that they may be harder for the FPMs relative to the UCL. Hoshika et al. [31] reported that the anterior bundle of the UCL could be interpreted as the part of the tendinous complex composed of FDS, FCU, and joint capsule. From this, it was considered that the UCL altered its hardness and became harder tissue with the FDS contraction as shown in this study. In addition, Davidson et al. [26] reported that the FCU is anatomically located just above and overlying the UCL. FCU contraction was considered to harden the FPMs relative to the UCL.

In previous studies, the verification of the function of the dynamic medial elbow stabilizers was limited to the evaluation of the medial elbow joint width [20,30]. To our knowledge, this study is the first to investigate changes in medial elbow joint parameters, including soft tissues, due to selective contraction of the forearm muscles which are dynamic medial elbow stabilizers. To summarize the results of this study, although the medial elbow joint width was little changed by contraction of FCU alone, contraction of the entire FPMs, as well as by FDS and PT working alone or combined, significantly decreased medial elbow joint width and stabilized against the elbow valgus load compared to resting. The UCL was significantly harder (stiffer) on contraction tasks involving FDS. FPMs were little changed when PT contracted in isolation but significantly hardened when FCU and FDS contracted alone or in combination. There was no significant difference in the ratio of the hardness UCL/FPMs between the eight conditions. However, there was a tendency towards a harder UCL relative to the FPMs during contraction involving FDS. Contractions composed of FCU and PT tended to harden the FPMs relative to the UCL. Based on the study by Hattori et al. [25], who reported that the harder the FPMs relative to the UCL, the less likely it is for UCL laxity to occur during repeated pitching. Our study found that the FCU and PT contraction exercises harden the FPMs relative to the UCL and decrease the contribution of the UCL against elbow valgus load, which may be helpful for UCL injury prevention. Therefore, selective training of FCU and PT in baseball players may be effective in UCL injury prevention. However, the benefits of this form of injury prevention need to be clarified in future longitudinal studies. On the other hand, since the combined contraction of FPMs, including FDS, has a high contribution to the stabilization of the medial elbow joint, it is likely that combined contraction training of FPMs, including FDS, may be effective in improving valgus instability in baseball players with significant valgus instability. This may be indicated for baseball players who have significant valgus instability and where the UCL rubs against the trochlea of the humerus.

This study has several limitations. First, the participants in this study were not baseball pitchers. It is possible that baseball pitchers who routinely pitch repeatedly and apply load to the medial elbow joint may show different results. This is a basic study targeting general college students, and the characteristics of baseball pitchers need further confirmation. Second, this study did not examine the medial elbow joint parameters after forearm muscle exercises. In this study, we wanted to compare multiple muscle contraction tasks, so we evaluated medial elbow joint parameters during contraction for each task only. Therefore, carry-over effects after these selective contraction activities on medial elbow joint parameters need further investigation. Third, this study only included males, and the characteristics of females are not yet known. The characteristics of females need to be verified in the future. In addition, this study covered the 20–24 age group, which is a narrow age range. Future studies should increase the sample size and validate the results in a wider age range. Finally, ultrasound measurements were performed in situ with selective contractions by the participants. Therefore, there was no blinding in the measurements.

## 5. Conclusions

Medial elbow joint width was little changed by contraction of FCU alone, but contractions of entire FPMs as well as FDS and PT alone or combination significantly decreased this width suggesting stabilization of the medial elbow against elbow valgus load compared to rest. The UCL was significantly harder (stiffer) on contraction tasks involving FDS. FPMs were little changed when PT contracted in isolation but significantly hardened when FCU and FDS contracted alone or in combination. There was no significant difference in the ratio of hardness of the UCL/FPMs between the resting condition and the seven selective muscle contraction conditions. However, the contractions involving FDS tended to harden the UCL relative to the FPMs, and the contractions consisting of FCU and PT tended to harden the FPMs relative to the UCL. FCU and PT activation may be effective in UCL injury prevention. These findings may have implications for injury prevention programs and encourage further work in this area.

## Figures and Tables

**Figure 1 healthcare-11-00586-f001:**
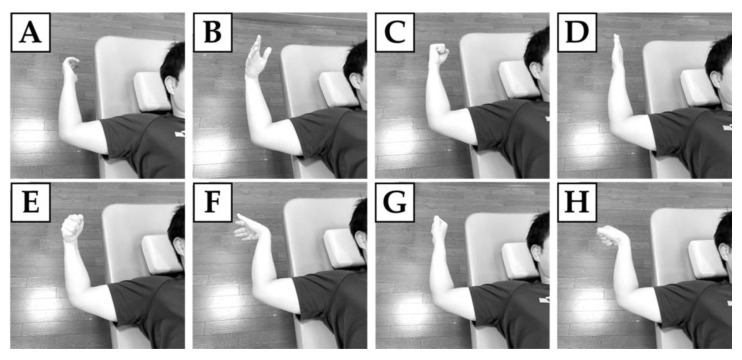
Muscle contraction tasks. (**A**) resting, (**B**) wrist ulnar flexion (contraction of FCU), (**C**) finger flexion (contraction of FDS), (**D**) forearm pronation (contraction of PT), (**E**) wrist ulnar flexion and finger flexion (contraction of FCU and FDS), (**F**) wrist ulnar flexion and forearm pronation (contraction of FCU and PT), (**G**) finger flexion and forearm pronation (contraction of FDS and PT), (**H**) wrist ulnar flexion, finger flexion, and forearm pronation (contraction of FCU, FDS, and PT). Each of the seven muscle contraction tasks (other than resting) was performed 3 times at maximum effort for 5 s. The order of these tasks was randomized.

**Figure 2 healthcare-11-00586-f002:**
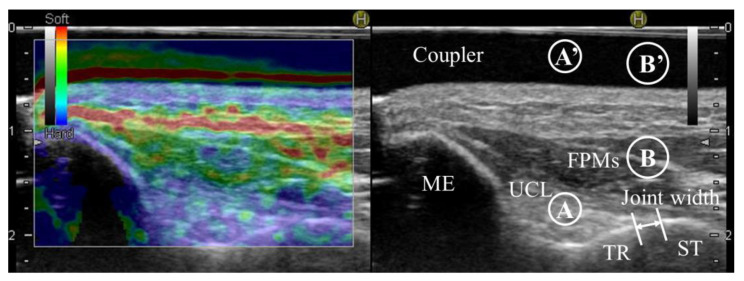
Evaluation of ultrasound images. The elastography image on the left is colored for each tissue hardness (red, soft; yellow to green, medium; blue, hard). The image on the right is a B-mode image. ME, medial epicondyle of humerus; TR, trochlea of humerus; ST, sublime tubercle of ulna; UCL, ulnar collateral ligament; FPMs, forearm flexor–pronator muscles; Coupler, coupler region; Joint width, ulnohumeral joint width; Circle A, target region of UCL; Circle A’, reference region of UCL (just above the UCL’s target region); Circle B, target region of FPMs; Circle B’, reference region of FPMs (just above the FPMs’ target region).

**Figure 3 healthcare-11-00586-f003:**
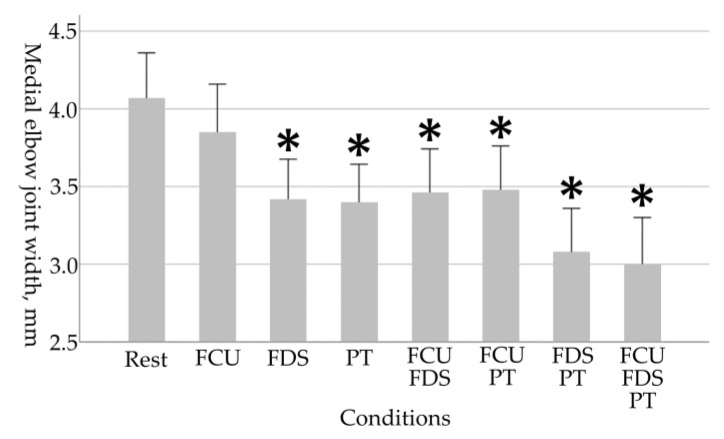
Changes in medial elbow joint width due to selective contraction of the FPMs. A higher value in medial elbow joint width indicates greater elbow valgus laxity. Asterisks indicate significant differences compared to the resting condition (*p* < 0.05). Error bars in the graph are standard errors. FPMs, forearm flexor–pronator muscles; FCU, flexor carpi ulnaris; FDS, flexor digitorum superficialis; PT, pronator teres.

**Figure 4 healthcare-11-00586-f004:**
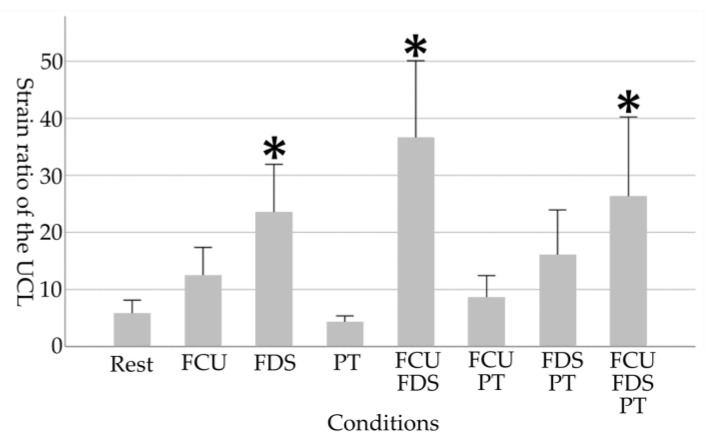
Changes in UCL due to selective contraction of the FPMs. A higher value in strain ratio indicates harder (stiffer) tissue. Asterisks indicate significant differences compared to the resting condition (*p* < 0.05). Error bars in the graph are standard errors. UCL, ulnar collateral ligament; FCU, flexor carpi ulnaris; FDS, flexor digitorum superficialis; PT, pronator teres.

**Figure 5 healthcare-11-00586-f005:**
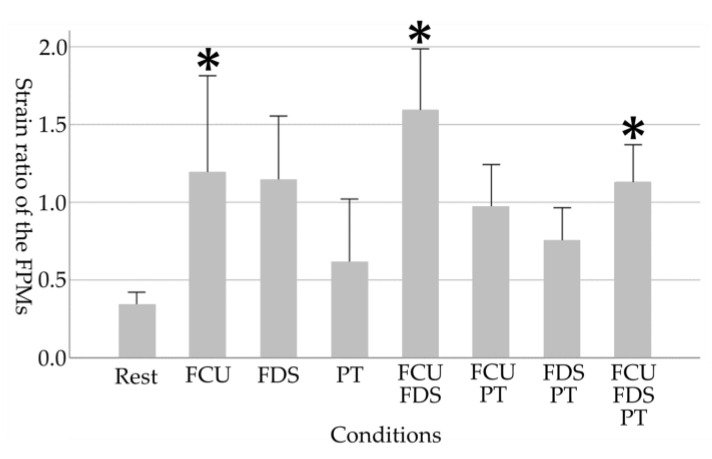
Changes in FPMs due to selective contraction of the FPMs. A higher value in strain ratio indicates harder (stiffer) tissue. Asterisks indicate significant differences compared to the resting condition (*p* < 0.05). Error bars in the graph are standard errors. FPMs, forearm flexor–pronator muscles; FCU, flexor carpi ulnaris; FDS, flexor digitorum superficialis; PT, pronator teres.

**Figure 6 healthcare-11-00586-f006:**
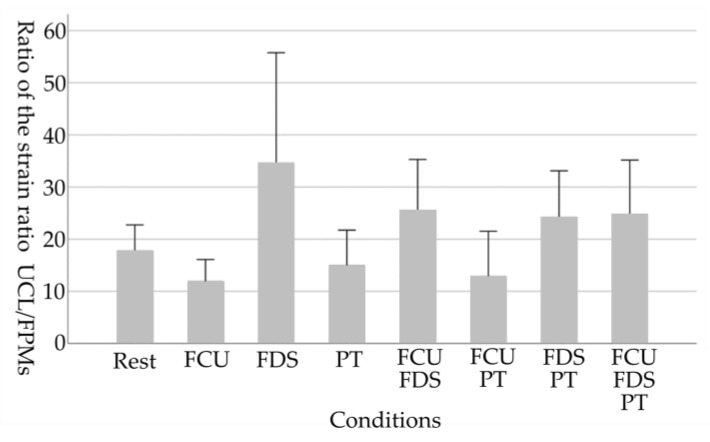
Changes in the ratio of the strain ratio UCL/FPMs due to selective contraction of the FPMs. A smaller value in the ratio of strain ratio UCL/FPMs indicates harder (stiffer) tissue of FPMs relative to that of UCL. Error bars in the graph are standard errors. UCL, ulnar collateral ligament; FPMs, forearm flexor–pronator muscles; FCU, flexor carpi ulnaris; FDS, flexor digitorum superficialis; PT, pronator teres.

**Table 1 healthcare-11-00586-t001:** Changes in medial elbow joint parameters due to selective contractions of the FPMs (n = 20).

	Rest	FCU	FDS	PT	FCU and FDS	FCU and PT	FDS and PT	FCU, FDS, and PT
Medial elbow joint width, mm	4.07 ± 0.65	3.85 ± 0.69	3.42 ± 0.58	3.40 ± 0.55	3.46 ± 0.63	3.48 ± 0.63	3.08 ± 0.63	3.00 ± 0.68
*p* value (vs. rest) ^1^		0.802	0.008	0.006	0.017	0.022	<0.001	<0.001
Strain ratio of the UCL	5.84 ± 5.12	12.43 ± 11.08	23.58 ± 18.68	4.30 ± 2.40	36.66 ± 30.04	8.57 ± 8.61	16.08 ± 17.59	26.36 ± 31.02
*p* value (vs. rest) ^1^		0.780	0.017	1.000	<0.001	0.997	0.346	0.004
Strain ratio of the FPMs	0.34 ± 0.18	1.20 ± 1.38	1.15 ± 0.91	0.62 ± 0.90	1.60 ± 0.88	0.97 ± 0.60	0.76 ± 0.46	1.13 ± 0.53
*p* value (vs. rest) ^1^		0.006	0.012	0.809	<0.001	0.076	0.418	0.014
UCL/FPMs	17.89 ± 10.84	12.02 ± 9.11	34.66 ± 47.21	15.11 ± 14.83	25.60 ± 21.66	12.99 ± 19.08	24.33 ± 19.64	24.84 ± 23.12
*p* value (vs. rest) ^1^		0.949	0.126	0.999	0.834	0.980	0.921	0.891

UCL, ulnar collateral ligament; FPMs, forearm flexor–pronator muscles; FCU, flexor carpi ulnaris; FDS, flexor digitorum superficialis; PT, pronator teres. *p* value (main effect): medial elbow joint width, <0.001 by one-way repeated-measures ANOVA; strain ratio of the UCL, <0.001 by Friedman’s test; strain ratio of the FPMs, <0.001 by Friedman’s test; UCL/FPMs, 0.013 by Friedman’s test. ^1^ Dunnett’s test.

## Data Availability

The data presented in this study are available on request from the corresponding author.

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
