# Peer review of "Changes in Medial Elbow Joint Parameters Due to Selective Contraction of the Forearm Flexor–Pronator Muscles"

_healthcare, 2023, doi:10.3390/healthcare11040586_

Round 1

Reviewer 1 Report

Dear Authors,

I would like to congratulate you on the conducted research - it is well conducted and precisely described. I have only minor suggestions that could improve this manuscript. The Table 1 is difficult to read in this orientation. Could you consider reworking it for different presentation? I understand, that it may be hard, but please check the possibilities. You should improve the quality of figures - they seems to be a bit blurred. I would also suggest to broaden the section of clinical implications in the discussion.

With regards,

The reviewer

Reviewer 2 Report

Thank you for the presentation of the work. 

We are struck by several things I would like to classify:

1- there is no difference between men and women or gender? you have not put it in your material and methodology. it is a compulsory factor to mention the gender of the subjects and if it influences the result.

2- we are not clear about the starting position and how the work was carried out. a photo or diagram of the positions and the material is necessary.

3- the sample is very young with an average age of 24 years, which makes the result conditioned even partly because the sample is very small.

4- What test did you use to find out the laterality or dominant side of the participants? how did you do it?

thank you 

Reviewer 3 Report

Nice study with possible applications in various healthcare areas, although a comparison between healthy elbows of men and women, of athletes versus non-athletes, of the dominant and non-dominant side is missing. Also, prior knowledge of the functional state of the structures in other variables other than the separation and elasticity of structures. 

It would be convenient to indicate in the title the specific characteristics of the elbows evaluated. Also, to reflect the sporting activity carried out by the sample subjects (if any). It is recommended the inclusion of a flow chart with the selection process of the dead. Was the experiment recorded?

Finally, authors are advised to include a "procedure" section right after participants. This section should accurately describe the steps performed during the applied method. Were the test subjects blinded?

Round 2

Reviewer 2 Report

good modification 

thank you for your work